# A biological switching valve evolved in the female of a sex-role reversed cave insect to receive multiple seminal packages

Kazunori Yoshizawa[1]*, Yoshitaka Kamimura[2], Charles Lienhard[3], Rodrigo L Ferreira[4], Alexander Blanke[5,6]

[1]Laboratory of Systematic Entomology, School of Agriculture, Hokkaido University, Sapporo, Japan; [2]Department of Biology, Keio University, Yokohama, Japan; [3]Natural History Museum of Geneva, Geneva, Switzerland; [4]Biology Department, Federal University of Lavras, Lavras, Brazil; [5]Institute for Zoology, University of Cologne, Zülpicher, Köln; [6]Medical and Biological Engineering Research Group, School of Engineering and Computer Science, University of Hull, Hull, United Kingdom

**Abstract** We report a functional switching valve within the female genitalia of the Brazilian cave insect *Neotrogla*. The valve complex is composed of two plate-like sclerites, a closure element, and in-and-outflow canals. Females have a penis-like intromittent organ to coercively anchor males and obtain voluminous semen. The semen is packed in a capsule, whose formation is initiated by seminal injection. It is not only used for fertilization but also consumed by the female as nutrition. The valve complex has two slots for insemination so that *Neotrogla* can continue mating while the first slot is occupied. In conjunction with the female penis, this switching valve is a morphological novelty enabling females to compete for seminal gifts in their nutrient-poor cave habitats through long copulation times and multiple seminal injections. The evolution of this switching valve may have been a prerequisite for the reversal of the intromittent organ in *Neotrogla*.
DOI: https://doi.org/10.7554/eLife.39563.001

*For correspondence:
psocid@res.agr.hokudai.ac.jp

**Competing interests:** The authors declare that no competing interests exist.

## Introduction

Many man-made engineering solutions have evolved already in insects. Such examples include hinges (flapping flight enabled by the wing base: **Brodsky, 1994**), on-off valves (spiracle openings to regulate airflow: **Chapman, 1998**), backflow valves (the bombardier beetle's defensive spray: **Arndt et al., 2015**), coiling mechanisms (genital tubes: **Matsumura et al., 2017a**) or catapult-like mechanisms (the legs of many jumping insects: **Burrows, 2013**). Some mechanisms that were formerly thought to be unique to human engineering have been discovered recently in insects: biological screws have been found in a beetle's leg (**van de Kamp et al., 2011**), and interacting gears were found in the jumping legs of planthoppers (**Burrows and Sutton, 2013**). The design and construction of such mechanisms on a micrometer scale is a challenging task in engineering (**Feinberg et al., 2001**). Therefore, studies of micron-scale biological structures can be rewarding as they illuminate construction principles in insects that could be applied to technical solutions in engineering (**Matsumura et al., 2017b**).

The genus *Neotrogla* (family Prionoglarididae) is a minute Brazilian cave insect belonging to the order Psocodea (booklice, barklice, and parasitic lice). This genus is of special evolutionary and morphological interest because of the reversal in its genital structures (**Yoshizawa et al., 2014**). The

**eLife digest** In dry caves of southeastern Brazil, live a group of insects named *Neotrogla* that are perhaps best known because the egg-producing females have penises while the sperm-producing males have vaginas. The sex roles of these Brazilian cave insects are also reversed: females compete over the males, who in turn are selective of their female partners. This sex role reversal likely evolved within *Neotrogla* because the males' semen represents a rich and reliable source of energy within a nutrient-poor cave environment. When females are not using semen to fertilize their eggs, they consume it. Yet, while other animals show sex role reversal, *Neotrogla* species alone have reversed sexual organs.

*Neotrogla* penises are spiky and may have evolved so that females can anchor themselves to males and then mate for prolonged periods. This would allow the females to stock up on the nutritious semen. Compared to their closest relatives, *Neotrogla* species can hold twice as much semen within their sperm storage organs. Scientists have speculated that a valve-like structure within this organ enables this extra storage by allowing the female to redirect semen into two separate chambers. But the organ's small size has made it difficult to determine its inner workings, and scientists have yet to discover a switching valve that serves such a purpose within the animal kingdom.

Yoshizawa et al. examined three *Neotrogla* species using advanced imaging technology and detected the first example of a biological switching valve. *Neotrogla* females can control this valve, switching the flow of semen between two slots. In this way, females can store two batches of semen at once. Seemingly exploiting this adaptation, the females' spiky penises help them restrain males until they have received multiple semen injections. Yoshizawa et al. therefore suggest the emergence of this valve within the sperm storage organ may have promoted the evolution of the female penis.

Along with giving insight into the lives of cave insects, these findings are of interest to engineers, who face challenges when constructing valves on a microscopic scale. The unique switching valve of female *Neotrogla* may one day inspire new man-made machinery that could advance a range of industries.

DOI: https://doi.org/10.7554/eLife.39563.002

females of *Neotrogla* have a penis-like intromittent organ (gynosome: *Figure 1A*), which is inserted to a male vagina-like genital cavity for copulation. During mating, the male injects liquid semen into the female's sperm storage organ (spermatheca) through the opening of the spermathcal duct at the tip of the female penis. Within the spermatheca, the injected semen then induces the formation of a hard capsule shell around itself (*Figure 1A,B*: *Wearing-Wilde (1995)*; *Yoshizawa et al., 2014*). Although there is only a single inlet spermathecal duct present, occasionally two seminal capsules are attached simultaneously to a plate-like structure on the spermatheca (termed 'spermathecal plate': [*Lienhard et al., 1893*; *Yoshizawa et al., 2014*]).

The semen within the capsule is used not only for fertilization but is also consumed by the female as nutrition. To compete for nutritious semen, the direction of sexual selection is reversed in *Neotrogla* (sex-role reversal: *Yoshizawa et al., 2014*). Each seminal capsule is voluminous (~0.05 mm$^3$, corresponding to ~300 ml scaled up to humans), and the duration of the copulation is very long (for 40 to 70 hr). In a closely related species lacking reversal of genital structure (*Lepinotus patruelis*, Trogiidae), the seminal transfer for forming a similarly voluminous seminal capsule is known to complete in 50 min (*Wearing-Wilde, 1995*). The female penis of *Neotrogla* bears a lot of spines, by which females anchor a male coercively during copulation. Therefore, females are obviously responsible for this very long copulation, probably to obtain more semen from a male (*Yoshizawa et al., 2014*). After consumption of the semen, the empty capsule is detached from the plate, which is kept within the spermatheca. Because females frequently have empty capsules within the spermathecal pouch (in an extreme case, up to nine empty capsules and two filled ones attached to the plate were observed: *Yoshizawa et al., 2014*), female *Neotrogla* can be considered polyandrous, which is apparently controlled actively by the female.

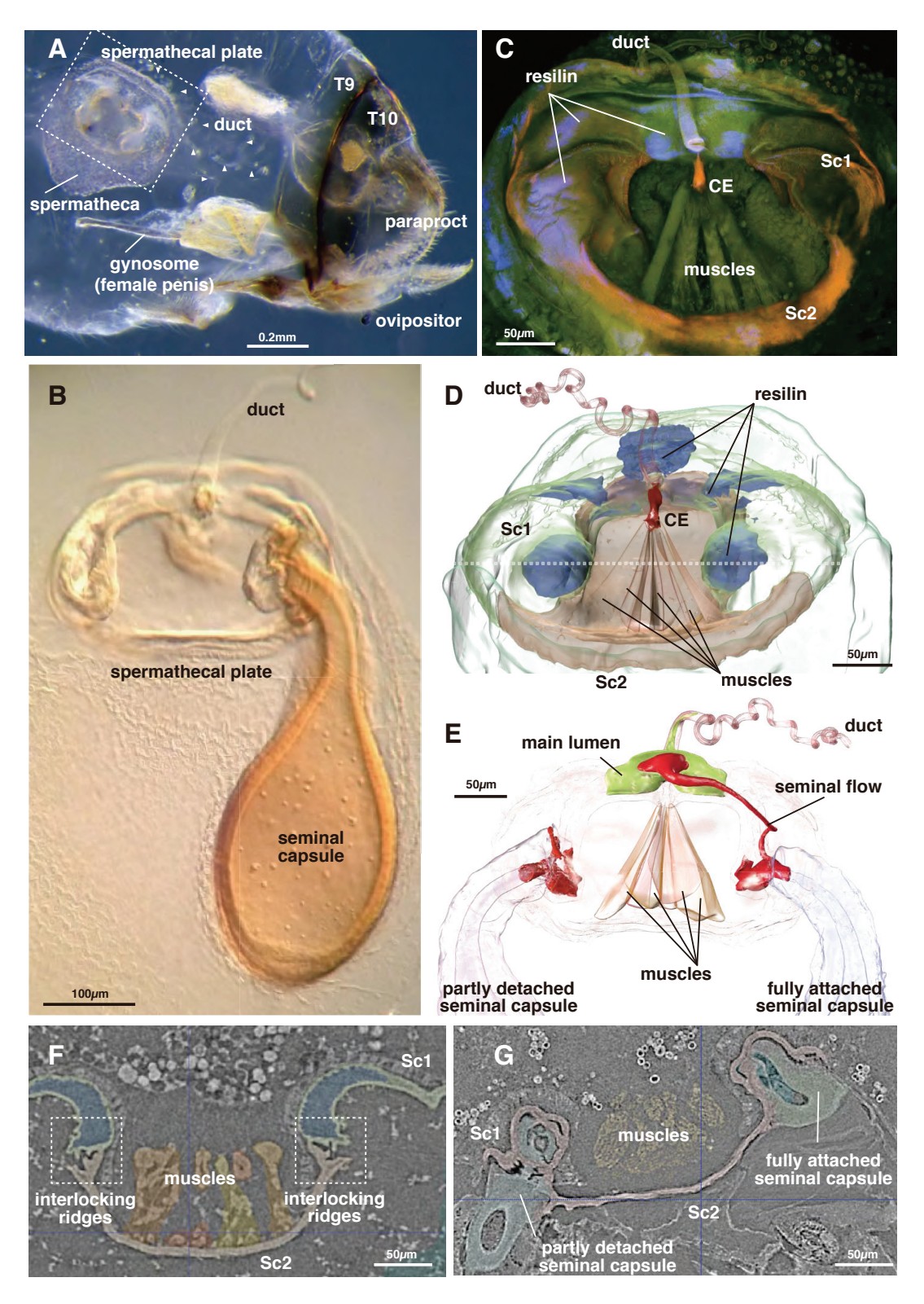

**Figure 1.** Morphology of the spermatheca and spermathecal plate of *Neotrogla*. (**A**) Whole abdomen of a virgin female showing the location of the mating system. T8 and 9 indicate tergites 8 and 9. (**B**) Light microscopy photograph of the spermathecal plate with a single seminal capsule. (**C**) CLSM image of the spermathecal plate. (**D**) 3D segmentation of the spermathecal plate with no seminal capsule. Dotted line indicates the section plane shown in F and G. (**E**) 3D segmentation of the spermathecal plate with two seminal capsules and showing seminal flow. (**F**) Attachment points for the

*Figure 1 continued on next page*

*Figure 1 continued*

seminal capsules in the virgin female. (G) The base of two seminal capsules in different stages of connection to the spermathecal plate (corresponding to the two seminal capsules in *Figure 1E*).

DOI: https://doi.org/10.7554/eLife.39563.003

Although several examples of sex-role reversed animals induced by seminal gifting have been reported, *Neotrogla* so far is the only example with a morphological reversal of its intromittent organs (*Kamimura and Yoshizawa, 2017*). This morphological reversal seems to require further upstream modifications of the genital system, such as the formation of the spermathecal plate, which was reported as an additional novelty tightly associated with the seminal gifting (*Lienhard et al., 1893*; *Yoshizawa et al., 2014*). However, the detailed morphology and function of this spermathecal plate, which presumably plays an important role in the evolution of the female penis, remained unclear due to the extremely small size of the involved subcomponents and their fragile spatial composition.

In this study, we investigated the structure of the spermathecal plate by using a combination of confocal laser scanning microscopy (CLSM) and high-resolution synchrotron microcomputed tomography (HR-μCT) to assess the functional morphology of sperm storage and control of seminal flow. We examined three species of *Neotrogla* (*N. brasiliensis*, *N. aurora* and *N. truncata*), the spermathecal morphology of which is practically identical (*Lienhard et al., 1893*). Based on the results, we discuss the evolutionary significance of this plate during the evolution of the reversed intromittent organs in *Neotrogla*.

## Results

The spermatheca of *Neotrogla* is in principle composed of an extensible pouch for storage of the seminal capsule, a spermathecal plate divided into two interconnected sclerites (body sclerites 1 and 2: Sc1 and 2 hereafter), and the spermathecal duct, which discharges into the spermathecal plate (*Figure 1CD*, *Video 1*). The terminology used for the following description is summarized in *Table 1*.

The switching valve mechanism is located at the spermathecal plate. Sc 2 is a bowl-shaped chitinous structure harboring a fan-like muscle (the actuator), which originates at its ventral part and attaches to a thumb-shaped controller/closure element (CE: *Figures 1CD* and *2A–D*). Given their attachment area, the force range of each muscle bundle is between 0.0758–0.6562 mN (assuming a standard intrinsic muscle force of 33 N/cm$^2$: *David et al., 2016*). The CE is located at the dorsal connection of Sc1 and 2, where the spermathecal duct opens into the pouch (*Figure 2A–D*).

In the virgin female, Sc 2 is connected to Sc 1 laterally through an interlock-like structure composed of several ridges on both sides that fit into each other (*Figure 1F*). This is the location where seminal capsules are formed during copulation (*Figure 1G*). Both Sc1 and Sc2 possess a system of patches of resilin, a rubber-like protein found in arthropod cuticles wherever potential energy is stored for spring-like motions or bending of structures. The region surrounding the CE contains extensive resilin patches (*Figures 1CD* and *2*). Two larger patches are located where the seminal capsules are attached so that this region can expand to harbor the base of the seminal capsule (*Figure 1DF*). Another resilin-rich region is located around the valve mechanism (*Figures 1CD* and *2*), where the resilin serves to passively keep the valve opening in the *closed* position. Opening of the valve can only occur through activation of the actuator muscle bundle. In the fixed material analyzed, the

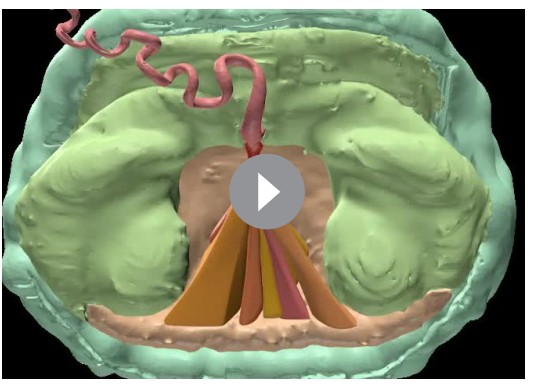

**Video 1.** 3D segmentation of the spermatheca and two seminal capsules attached to the spermathecal plate (see *Figure 1E*).

DOI: https://doi.org/10.7554/eLife.39563.005

**Table 1.** Valve terminology used in the text

| Actuator | Device used to operate a valve using electric, pneumatic or hydraulic means |
|---|---|
| Body | The principal pressure-containing part of a valve in which the closure element and seats are located |
| Closure element (CE) | The moving part of a valve, positioned in the flow stream, that controls the flow through the valve, for example wedge, plug, clapper, ball |
| Controller | A device that directs the flow of a valve |
| End connection | The type of connection supplied on the ends of a valve that allows it to be connected to piping — may be a welded end, flanged end, threaded or socket weld |
| Pennation angle | The oblique attachment of single muscle fascicles to the CE. It was measured as the angle between the outermost fascicles in a given muscle bundle |

DOI: https://doi.org/10.7554/eLife.39563.004

muscle bundles are contracted due to the fixation process. Therefore, the switching valve is in an *opened* position in *Figures 1–2*.

The seminal fluid entering the main lumen of the duct can be directed into the left or right channel depending on the position of the CE. If the left part of the actuator muscle is activated, the CE is moved to the left, so that a channel on the right side opens, which then allows for seminal flow into the right capsule (*Figure 2E*). A small lateral extension at the distal end of the CE serves to close the opposite channel during the opening of the other channel (*Figure 2DE*). If the right part of the actuator is activated, the process is executed in reverse. In a female fixed during copula, seminal flow from the duct opening lumen toward one of two seminal capsules was clearly observed (*Figure 1E*). Seminal flow can thus be directed by the female through differential muscle activation that moves the CE.

Measurements of each structure and the estimated power produced by the discernible muscle bundles mentioned above are summarized in *Table 2*.

## Discussion

The present analyses show that females of *Neotrogla* use a special mechanism to actively control the direction of seminal flow (*Figures 1E* and *2A–D*). This biological switching valve allows females to receive two seminal packages (i.e., more nutrition) from the same or different males within a short time span. In particular, the switching valve allows to receive a second seminal capsule while the first one is consumed. A similar but less sclerotized structure can be observed in close relatives of *Neotrogla* (*Sensitibilla* and *Afrotrogla*, all belonging to the tribe Sensitibillini), suggesting that the structure probably originated in their common ancestor (*Lienhard, 1893*; *Lienhard et al., 1893*). *Neotrogla* is distributed in South America, whereas *Sensitibilla* and *Afrotrogla* are distributed in southern Africa. Therefore, the origin of this switching valve dates to at least the break-up of the two continents, over 100 million years ago (*Seton et al., 2012*). In other psocodeans, no sclerite or muscle corresponding to those of the spermathecal plate elements have been observed (*Badonnel, 1934*; *Klier, 1956*; *Wearing-Wilde, 1995*). The spermathecal plate thus clearly represents an evolutionary novelty (*Müller and Wagner, 2003*).

The mechanism of flow control and redirection within this biological switching valve is fundamentally different from that in man-made switching valves, which are used, for example in the oil and gas industry. Technical switching valves use the active rotation of tube or ball elements with various openings to redirect the flow and both closing as well as opening involve an active movement of the valve. In contrast, the observed biological switching valve involves a passive movement component. The seminal flow injected by a male is redirected by a differential inclination of the CE (*Figure 2E*). This inclination is due to the fan-like geometry of the actuator muscle whose parts can pull the CE in the preferred direction to work against the passive closing forces generated by the resilin patches around the valve complex. This muscle-closure element configuration could be advantageous since

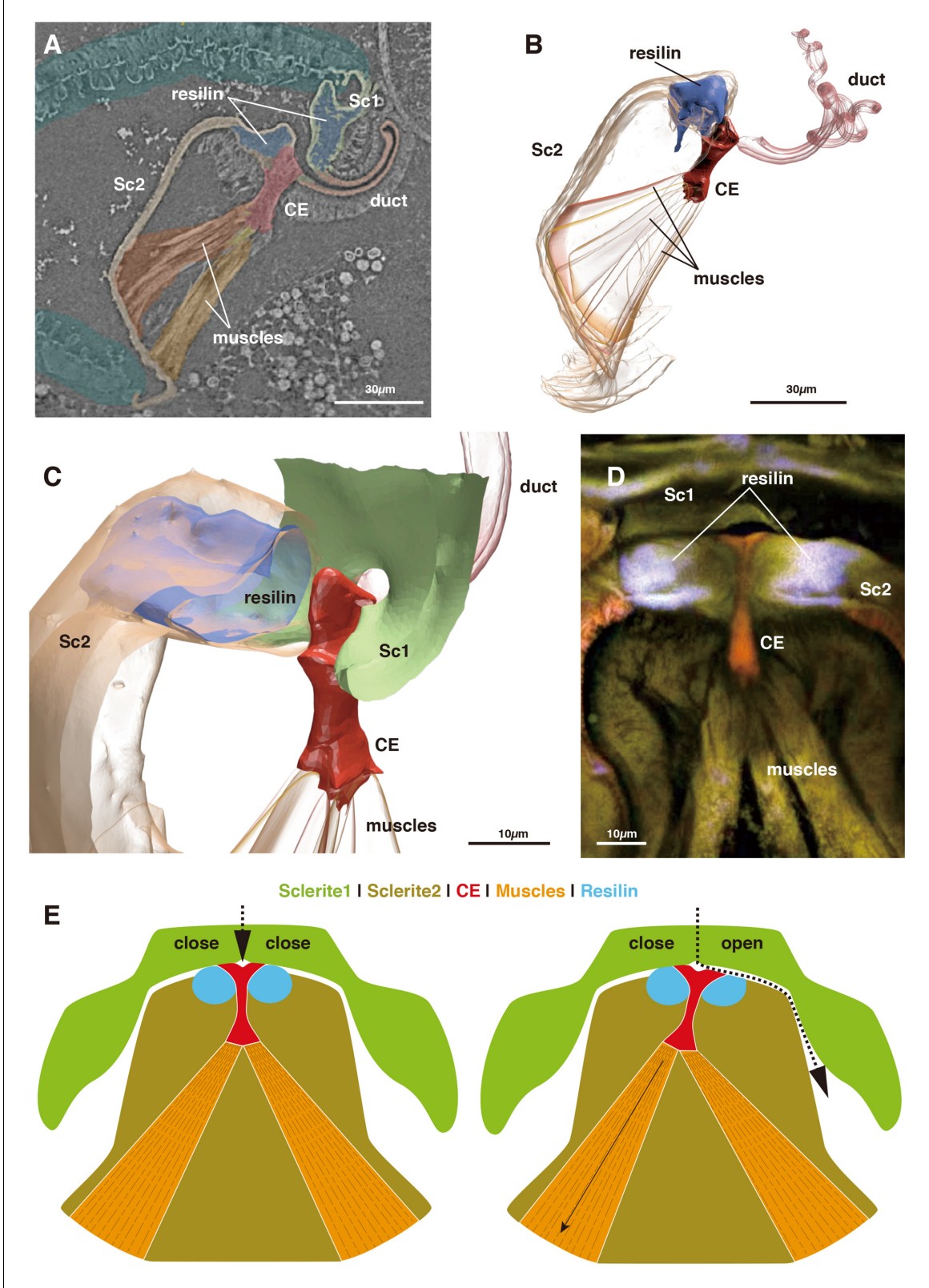

**Figure 2.** Morphology of the closure element (CE) and its associated structures. (**A**) Muscle attachment to CE. (**B**) 3D segmentation of *Figure 2A*. (**C**) Detail of CE. (**D**) CLSM image of CE and neighboring structures. (**E**) Schematic illustration of the function of the switching valve system in closed (left) and opened (right) conditions.

DOI: https://doi.org/10.7554/eLife.39563.006

**Table 2.** Measurements of each component of the spermathecal plate.

| Object | Pennation angle | Attachment area of muscle [cm2] | Muscle strength [mN] | Length [mm] | Volume [μm3] | Mass [μg] |
|---|---|---|---|---|---|---|
| Muscle m1 | 34.16 | 0.0000199 | 0.6562 | 0.1 | | |
| Muscle m2 | 30.18 | 0.0000191 | 0.6319 | 0.08 | | |
| Muscle m3 | 15.59 | 0.0000023 | 0.0758 | 0.09 | | |
| Muscle m4 | 44.55 | 0.0000133 | 0.4374 | 0.09 | | |
| Muscle m5 | 12.26 | 0.0000033 | 0.1096 | 0.09 | | |
| Muscle m6 | 32.75 | 0.0000073 | 0.2394 | 0.11 | | |
| Sum of muscle strength | | | 2.1503 | | | |
| Closure element | | | | 0.03 | | |
| Spermathecal pouch | | | | | 1847443779 | 0.0021246 |
| Body sclerite 1 | | | | | 199709300 | 0.0002297 |
| Body sclerite 2 | | | | | 101434410 | 0.0001166 |

DOI: https://doi.org/10.7554/eLife.39563.007

each muscle bundle pulls in an optimal direction to incline the CE into one of the two opening positions. Additionally, there is no need for lubrication since the moveable parts do not move against each other. Compared to technical switching valves, this design is advantageous to some extent because the muscle forces are not redirected via a lever arm.

The presence of a spermathecal plate with a valve function to control seminal flow may further refine our understanding of the causes and consequences of the reversed sexual selection in this genus. Female-female competition for males (sex-role reversal) to receive nutritious seminal substances is considered as the most important factor driving the evolution of the female penis (*Yoshizawa et al., 2014*). Among all animals with known sex-role reversal, *Neotrogla* is the only example in which a female penis evolved (*Kamimura and Yoshizawa, 2017*). In a close relative of *Neotrogla,* the barklouse species *Lepinotus patruelis* (*Wearing-Wilde, 1996*), the sex-roles are also reversed, but the species possesses normal genital structures. Therefore, it is very likely that, in addition to the sex-role reversal, there is at least one other key factor that enabled the evolution of a female penis. Although females of all three known genera of Sensitibillini possess a spermathecal plate, females of *Sensitibilla* do not have a penis-like organ (*Lienhard, 1893*; *Lienhard et al., 1893*). This strongly suggests that the evolution of the spermathecal plate, possibly including a switching valve, preceded the evolution of the female penis.

The spermathcal plate has two slots available for insemination (*Figure 1E*). In *Neotrogla* and the related species (*Lepinotus*), content of the capsule is digested as nutrition during the seminal capsule being attached to the spermathecal plate (the transparent capsule shown in *Figure 1B* is a digested and empty one: *Yoshizawa et al., 2014*). Therefore, if there is only one slot for insemination, as in the spermatheca of *Lepinotus*, females cannot receive another capsule while digesting one. With the switching valve, the female *Neotrogla* (and possibly *Sensitibilla* and *Afrotrogla*) can selectively use one of two slots for insemination, with leaving the other slot empty. This enables the females to immediately receive an additional seminal package from the same or other males by using the empty slot. Males are predicted to prudently allocate limited resources, such as nutritious seminal gifts and sperm, to multiple females, especially when operational sex ratio is biased to females, rendering female-female competition for male-derived nuptial gifts (i.e, propensity for multiple mating) more intense (*Abe and Kamimura, 2015*). The male-holding organ (spiny female penis) and female-induced long copulation durations of 40 – 70 hr (compared to just ~50 min for formation of one seminal capsule in close relatives) in *Neotrogla* (*Yoshizawa et al., 2014*) thus can be considered as exaggerated adaptations for such escalated competition for nuptial gifts in this group of insects inhabiting highly oligotrophic cave habitats (*Lienhard and Ferreira, 2013*; *Lienhard and Ferreira, 2015*; *Yoshizawa et al., 2014*).

The condition of having two freshly deposited spermatophores at once is comparable to that in multiple sperm storage organs reported for females of some animal groups, such as dung flies, *Drosophila*, or tephritid fruit flies (*Ward, 1993*; *Pitnick et al., 1999*; *Twig and Yuval, 2005*). Although

theory predicts that having multiple sperm stores can be a powerful mechanism for choosing sperm (*Hellriegel and Ward, 1998*), evidence is scarce for a gain in fitness by actively selecting for particular sperm from among multiple mates (e.g., *Demont et al., 2012*; *Schäfer et al., 2013*). In addition, in the case of *Neotrogla* and related barklice (*Lepinotus patruelis*), the content of the seminal capsule is digested quite rapidly if not used for fertilization (*Wearing-Wilde, 1995*; *Yoshizawa et al., 2014*). Therefore, the switching valve system reported here likely represents an adaptation for direct benefits (i.e., for obtaining more nutrients) rather than for genetic benefits (i.e., for choosing sperm from high-quality males).

## Materials and methods

Three species of *Neotrogla* were examined. We detected little interspecific variation in the basic mechanism of the spermathecal plate.

A virgin female of *Neotrogla brasiliensis* (Caboclo Cave, Januária, Minas Gerais, Brazil, 12. iii. 2016: *Figures 1DF* and *2A–C*: voucher ID S8KY03) and a copulating pair of *N. truncata* (Toca dos Ossos Cave, Ourolândia, Bahia, Brazil, 14. i. 2013: *Figure 1EG*: voucher ID S8KY69: full shape data provided as *Video 1*) were used for μCT examination (http://dx.doi.org/10.6084/m9.figshare.6741857). *Neotrogla brasiliensis* was fixed with FAA solution (formaldehyde-acetic acid-alcohol) and *N. truncata* was fixed with 80% ethanol. Both samples were then stored in 80% ethanol. Dehydration was conducted in ascending order with 80 – 100% ethanol before drying them at the critical point (EM CPD300, Leica, Wetzlar, Germany) to remove water without serious organ shrinkage. Samples were then scanned using synchrotron microcomputed tomography at the BL47XU (*Uesugi and Hoshino, 2012*) beamline of the Super Photon ring-8 GeV (SPring-8; Hyogo, Japan) using a stable beam energy of 8 keV in absorption-contrast mode. The tomography system consists of a full-field X-ray microscope with Fresnel zone plate optics (*Uesugi et al., 2017*). The FOV and effective pixel size are 0.11 mm$^2$ and 0.0826 μm$^2$, respectively. We used semiautomatic segmentation algorithms based on gray-value differences in the software ITK-SNAP (*Yushkevich et al., 2006*) to obtain 3D representations of the genitalia of *Neotrogla*. Rendering of the mesh objects was carried out using the software BLENDER (blender.org). Objects were imported as stl files, surface meshes were slightly smoothed, and the number of vertices were reduced without altering the 3D geometry. No further processing was applied. All measurements were carried out in BLENDER.

A virgin female of *N. brasiliensis* (*Figures 1C* and *2D*: voucher ID: CLKY1) was also used for confocal laser scanning microscope (CLSM) observation (Leica TCS SP5, Wetzlar, Germany). The spermathecal plate was removed and mounted on a glass slide with glycerol. We used an excitation wavelength of 488 nm and an emission wavelength of 510 – 680 nm, detected using two channels and visualized separately with two pseudocolors (510 – 580 nm = green; 580 – 680 nm = red). To visualize resilin, we used an excitation wavelength of 405 nm and an emission wavelength of 420 – 480 nm, detected on one channel and represented with a blue pseudocolor.

A virgin female of *N. aurora* (Gruta Couve-Flor cave; Aurora do Tocantins, Tocantins, Brazil, 7. i. 2009) was used to take the whole-abdomen photo shown in *Figure 1A*. The abdomen was removed from a fixed specimen and soaked in Proteinase K at 45°C overnight and stored in glycerol. Photographs were taken with an Olympus E-M5 digital camera attached to an Olympus SZX16 binocular microscope (Tokyo, Japan). Partially focused pictures were combined using Helicon Focus (Helicon Soft Ltd., http://www.heliconsoft.com) to obtain images with a high depth of field. The holotype female of *N. truncata* (*Lienhard et al., 1893*) was used for photographing the spermathecal plate shown in *Figure 1B*. Photographs were taken with an Olympus E-M5 attached to a Zeiss Axiophot compound light microscope (Oberkochen, Germany).

## Acknowledgements

We thank Marconi Souza-Silva for support in the field, Kentaro Uesugi for his support with the μCT imaging at SPring-8, Masanori Yasui and Naoki Ogawa for supporting CLSM imaging. We also thank Liuqui Gu and two anonymous reviewers for their very constructive comments. Research at SPring-8 was approved through project numbers 2016A1269 (leader: Ryuichiro Machida) and 2017B1712 (leader: Naoki Ogawa). Japan Society for the Promotion of Science, 15H04409, Kazunori Yoshizawa and Yoshitaka Kamimura. Conselho Nacional de Desenvolvimento Cientifico e Tecnológico, 304682/

2014 – 4, Rodrigo Ferreira. European Research Council, 754290 ('Mech-Evo-Insect'), Alexander Blanke

## Additional information

### Funding

| Funder | Grant reference number | Author |
|---|---|---|
| Japan Society for the Promotion of Science | 15H04409 | Kazunori Yoshizawa Yoshitaka Kamimura |
| Conselho Nacional de Desenvolvimento Científico e Tecnológico | 304682/2014-4 | Rodrigo L Ferreira |
| European Research Council | 754290 | Alexander Blanke |

The funders had no role in study design, data collection and interpretation, or the decision to submit the work for publication.

### Author contributions

Kazunori Yoshizawa, Conceptualization, Data curation, Formal analysis, Funding acquisition, Investigation, Visualization, Writing—original draft, Writing—review and editing; Yoshitaka Kamimura, Conceptualization, Funding acquisition, Investigation, Writing—original draft, Writing—review and editing; Charles Lienhard, Investigation, Writing—review and editing; Rodrigo L Ferreira, Funding acquisition, Investigation, Writing—review and editing; Alexander Blanke, Data curation, Formal analysis, Funding acquisition, Investigation, Visualization, Writing—original draft, Writing—review and editing

### Author ORCIDs

Kazunori Yoshizawa (iD) http://orcid.org/0000-0001-6170-4296
Alexander Blanke (iD) http://orcid.org/0000-0003-4385-6039

### Decision letter and Author response

Decision letter https://doi.org/10.7554/eLife.39563.012
Author response https://doi.org/10.7554/eLife.39563.013

## Additional files

### Supplementary files

• Transparent reporting form
DOI: https://doi.org/10.7554/eLife.39563.008

### Data availability

Scanned images of *Neotrogla brasiliensis* have been deposited to FigShare and are available at: http://dx.doi.org/10.6084/m9.figshare.6741857.

The following dataset was generated:

| Author(s) | Year | Dataset title | Dataset URL | Database and Identifier |
|---|---|---|---|---|
| Kazunori Yoshizawa, Yoshitaka Kamimura, Rodrigo L Ferreira, Charles Lienhard, Alexander Blanke | 2018 | Biological Switching Valve | http://dx.doi.org/10.6084/m9.figshare.6741857 | Figshare, 10.6084/m9.figshare.6741857 |

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
