## [Decision Letter]

Thank you for submitting your article "A biological switching valve evolved in the female of a sex-role reversed cave insect to receive more seminal gifts" for consideration by *eLife*. Your article has been reviewed by Ian Baldwin as the Senior Editor, a Reviewing Editor, and three reviewers. The following individuals involved in review of your submission have agreed to reveal their identity: Liuqi Gu (Reviewer #3).

The reviewers have discussed the reviews with one another and the Reviewing Editor has drafted this decision to help you prepare a revised submission.

The three reviews were all in general agreement that the work provided valuable insights into the valve, but that the work required more careful discussion and development of the biological context regarding a number of key points, which are articulated in the reviews below. The reviews are provided in full below and we would like to see a point-by-point description of how you have addressed the concerns when you submit the revision. While the vast majority of the reviews can be addressed without new data, the last major point of the first review may require new data to address fully.

*Reviewer #1:*

Authors have detailed the mechanism for sperm storage in *Neotrogla*. The work they have done is of high quality and given the unique nature of the female intermittent organ in this species, is interesting to biologists interested in sexual selection. The valve described, although relatively simple, does not seem to be used in engineered products (though I am definitely not an expert here), and thus could serve as inspiration for the design of new valves. Although the work here is sound, I do have some concerns, detailed below.

My major issue with the paper is in the explanation, which is light, and interpretation which sometimes over extrapolates from what the authors have found. In particular, (as stated in the Title) the author claim that the switching valve has evolved to receive more seminal gifts. I am not sure I understand the logic here. I think the fact the female has two capsules allows for more sperm storage, I don't think having a switching valve per se means a female can receive more sperm (i.e. if there was no valve, the sperm capsules would fill equally from two matings). Given the clear complexity of the valve described, evolving a multiple / a larger spermatheca seems a much simpler solution to evolve if the only factor driving its evolution is to receive more sperm. This is a key point as the authors interpret their findings in this light, while dismissing other explanations (e.g. Discussion section), and so should be considered more carefully.

The authors also state that females also have 'detached' capsules. Is the sperm in these capsules used for fertilization? Or is only sperm in the attached capsules used for fertilization? This would have consequences for the interpretation of their evolution. Perhaps an advantage of the switching valve would be, that following detachment of a capsule from the spermathecal plate, there is lag time in producing a new capsule (?), and thus by having two there will be more chance of having a capsule ready if they encounter a male soon after one has been detached?

In addition, the paper suggests that a single seminal capsule used in a single mating, however I don't think the authors actually show this (i.e. could be possible that a female switches which sperm capsule she is filling during the long mating period?).

*Reviewer #2:*

This is a very interesting study on the fascinating booklice genus *Neotrogla*. These tiny insects are adapted to extreme cave habitats and have evolved exceptional genital structures associated with a sex-role reversal. Males produce large, nutritious spermatophores that the females store in their spermathecae for fertilisation as well as for nutrition. Sperm are apparently taken up by females, which possess a penis like intromitting structure that also secures a firm attachment of the male to the female. Copulations can take several days and the general sexual biology suggests that females compete for nutritious seminal capsules and coerce males into lengthy copulations and perhaps into producing additional sperm packages. Little is known about the ecology and behaviour of the species so that the costs of producing seminal capsules as well as the potential benefits for females to receive multiple capsules must remain speculative.

The current study is concerned with a special valve that females can use to direct the sperm flow. The morphology of female genitals is investigated using μCT and 3D reconstruction as well as confocal laser scanning microscopy.

I have no major concerns and only editorial suggestions. However, not being a morphologist, I had some trouble understanding what happens during copulation. In a previous study, I found highly relevant additional information that was indeed crucial for my comprehension of the highly peculiar sex in this animal group. To the benefit of a non-specialised readership I would therefore advise to add more detail and less technical language about the copulation procedure.

I appreciate that word counts are limited but perhaps some of the information could be included into the figure legends if there is no space in the text. To my taste, parts about valve technology could be replaced to some extend with some more details on sexual biology.

I am fascinated by these animals and am looking forward to additional future insights into their reproductive biology.

*Reviewer #3:*

The authors present an interesting story about a biological valve in female reproductive system of a cave insect. The manuscript is well composed with clear organization and good writing. The description is on point and the figures are of decent quality. Therefore, I consider its publication in *eLife* to be appropriate, with no additional work required.

Although, I wish the authors may address the following concern:

Three species of *Neotrogla* were examined in this study, however, this is not mentioned except in the Materials and methods section, which follows after Discussion section. I think not only it's necessary to describe this information in the main text (Abstract, Introduction, Results section and Discussion section), but also the authors should describe any nuances in morphological difference among the three species examined, if any.

---

## [Author Response]

The three reviews were all in general agreement that the work provided valuable insights into the valve, but that the work required more careful discussion and development of the biological context regarding a number of key points, which are articulated in the reviews below. The reviews are provided in full below and we would like to see a point-by-point description of how you have addressed the concerns when you submit the revision. While the vast majority of the reviews can be addressed without new data, the last major point of the first review may require new data to address fully.Reviewer #1:Authors have detailed the mechanism for sperm storage in Neotrogla. The work they have done is of high quality and given the unique nature of the female intermittent organ in this species, is interesting to biologists interested in sexual selection. The valve described, although relatively simple, does not seem to be used in engineered products (though I am defiantly not an expert here), and thus could serve as inspiration for the design of new valves. Although the work here is sound, I do have some concerns, detailed below.

Thank you very much for your interests and constructive comments to our paper.

My major issue with the paper is in the explanation, which is light, and interpretation which sometimes over extrapolates from what the authors have found.

We provide now more detailed explanations and toned down some over extrapolations. All of the following major concerns are rooted from our poor explanations for the formation of the seminal capsule. Therefore, our replies are given at the end of the three comments given from reviewer 1. Please also refer to the new main text of the manuscript for more details.

In particular, (as stated in the Title) the author claim that the switching valve has evolved to receive more seminal gifts. I am not sure I understand the logic here. I think the fact the female has two capsules allows for more sperm storage, I don't think having a switching valve per se means a female can receive more sperm (i.e. if there was no valve, the sperm capsules would fill equally from two matings). Given the clear complexity of the valve described, evolving a multiple / a larger spermatheca seems a much simpler solution to evolve if the only factor driving its evolution is to receive more sperm. This is a key point as the authors interpret their findings in this light, while dismissing other explanations (e.g. Discussion section), and so should be considered more carefully.The authors also state that females also have 'detached' capsules. Is the sperm in these capsules used for fertilization? Or is only sperm in the attached capsules used for fertilization? This would have consequences for the interpretation of their evolution. Perhaps an advantage of the switching valve would be, that following detachment of a capsule from the spermathecal plate, there is lag time in producing a new capsule (?), and thus by having two there will be more chance of having a capsule ready if they encounter a male soon after one has been detached?In addition, the paper suggests that a single seminal capsule used in a single mating, however I don't think the authors actually show this (i.e. could be possible that a female switches which sperm capsule she is filling during the long mating period?).

All of the above major concerns raised by reviewer 1 are caused by our poor explanation about how the seminal capsule is formed. Our and previous observations showed that capsules are formed within the females’ sperm storage system during copulation. Those observations show that the formation of the capsule is initiated by seminal transfer. Empty capsules are not prepared by the females as the reviewer might have assumed (again this is due to our previous poor explanations which have now been modified). The capsule formation process is least understood (see below), but probably initiated either by a substance provided by the male or by a protective response of the female against the semen (known as “melanisation”). Alternatively, capsule formation could be a result of a chemical reaction between the semen and spermathecal fluid. Because the female spermatheca does not have any muscles or structures enabling sucking up semen, the seminal transfer is apparently caused by an active infusion of the male, as is the case for the majority of animals including insects. When the semen was not used for fertilization, the content of the seminal capsule was consumed by females while the capsule was attached to the spermathecal plate. After consumption of the content of a capsule, the empty capsule detached from the plate before or during the next copulation. The detached capsule is kept within the spermatheca and is never dissolved.

The above knowledge on the seminal capsule formation was added in the revised manuscript (Introduction and Discussion section), which we hope remove the readers' confusion about the formation of the seminal capsule and helps to understand the function of the switching valve. In contrast, the detailed process of the formation of the sperm capsule is less well understood. One possible explanation, although not mentioned previously, is that the formation of the capsule shell can be the result of a chemical reaction between the contents of the semen and the spermathecal fluid (like a reaction between a drop of sodium alginate solution put into calcium chloride solution, which provides solid outer shell containing liquid sodium alginate solution). The size of all the sperm capsules is fairly constant possibly because it is: (1) limited by the amount of semen males can produce, or (2) constrained by the balance between the speed of seminal flow and speed of chemical reaction for formation of the capsule shell. In either case (or combination of both), having two openings WITHOUT a switching valve would cause formation of two half-sized seminal capsules which would occupy both plate slots. This would in turn not increase the female's benefit since multiple matings are not possible while previous sperm capsules are consumed or dissolved. We think this explanation, although likely, is too speculative to discuss in the paper.

Having multiple spermathecae can also be an alternative option to obtain more semen. As also mentioned by reviewer 2 and in our paper, this option is employed by some insects and spiders. In these animals, liquid semen directly fills the spermatheca. In contrast, the evolutionary origin of a hard seminal capsule apparently preceded the evolution of the switching valve in Sensitibillini (formation of the sperm capsule is common in the outgroups). Such differences might have been a factor for the evolution of the switching valve, but it is also too speculative to discuss in the paper.

The main purpose of our paper is to report and describe the biological switching valve and discuss its potential relationship for the evolution of the female penis. Ultimate factors for the origin of the switching valve deserve attention by future investigations and it would be premature to discuss in the present paper. We therefore also modified the title to tone down the over extrapolations about the ultimate factor for the origin of the switching valve.

Reviewer #2:This is a very interesting study on the fascinating booklice genus Neotrogla. These tiny insects are adapted to extreme cave habitats and have evolved exceptional genital structures associated with a sex-role reversal. Males produce large, nutritious spermatophores that the females store in their spermathecae for fertilisation as well as for nutrition. Sperm are apparently taken up by females, which possess a penis like intromitting structure that also secures a firm attachment of the male to the female. Copulations can take several days and the general sexual biology suggests that females compete for nutritious seminal capsules and coerce males into lengthy copulations and perhaps into producing additional sperm packages. Little is known about the ecology and behaviour of the species so that the costs of producing seminal capsules as well as the potential benefits for females to receive multiple capsules must remain speculative.The current study is concerned with a special valve that females can use to direct the sperm flow. The morphology of female genitals is investigated using μCT and 3D reconstruction as well as confocal laser scanning microscopy.

Thank you very much for your very positive reaction and useful comments to improve the manuscript. Some of the concerns raised by reviewer 2 are in common with those by reviewer 1. Please also see the reply to reviewer 1, in which detailed explanation about the formation process of the sperm capsule is given.

I have no major concerns and only editorial suggestions. However, not being a morphologist, I had some trouble understanding what happens during copulation. In a previous study, I found highly relevant additional information that was indeed crucial for my comprehension of the highly peculiar sex in this animal group. To the benefit of a non-specialised readership I would therefore advise to add more detail and less technical language about the copulation procedure.I appreciate that word counts are limited but perhaps some of the information could be included into the figure legends if there is no space in the text. To my taste, parts about valve technology could be replaced to some extend with some more details on sexual biology.I am fascinated by these animals and am looking forward to additional future insights into their reproductive biology.

We now provided more detailed explanation about the mating biology and sex-role reversals in *Neotrogla* in the Introduction and Discussion section.

Reviewer #3:The authors present an interesting story about a biological valve in female reproductive system of a cave insect. The manuscript is well composed with clear organization and good writing. The description is on point and the figures are of decent quality. Therefore, I consider its publication in eLife to be appropriate, with no additional work required.

Many thanks for your interests and positive reaction.

Although, I wish the authors may address the following concern:Three species of Neotrogla were examined in this study, however, this is not mentioned except in the Materials and methods section, which follows after Discussion section. I think not only it's necessary to describe this information in the main text (Abstract, Introduction, Results section and Discussion section), but also the authors should describe any nuances in morphological difference among the three species examined, if any.

At the end of the Introduction, we added the information that the present study is based on the examination of three species. No notable morphological differences among these species can be detected in our examination, which was also noted at the beginning of the Material and methods section.